# Novel NADPH Oxidase-2 Inhibitors as Potential Anti-Inflammatory and Neuroprotective Agents

**DOI:** 10.3390/antiox12091660

**Published:** 2023-08-23

**Authors:** Matea Juric, Varun Rawat, Radhika Amaradhi, Jacek Zielonka, Thota Ganesh

**Affiliations:** 1Department of Biophysics, Medical College of Wisconsin, Milwaukee, WI 53226, USA; mjuric@mcw.edu; 2Department of Pharmacology and Chemical Biology, Emory University School of Medicine, Atlanta, GA 30322, USA; varun.rawat@emory.edu (V.R.); radhika.amaradhi@utsa.edu (R.A.)

**Keywords:** NADPH oxidase, inflammation, neurodegeneration, inhibitors, neuroprotection, superoxide, hydrogen peroxide

## Abstract

A family of seven NADPH oxidase enzymes (Nox1-5, Duox1-2) has been implicated in a variety of diseases, including inflammatory lung diseases, neurodegenerative diseases, cardiovascular diseases, and cancer. Here, we report the results of our studies aimed at developing novel brain-permeable Nox2 inhibitors with potential application as neuroprotective agents. Using cell-based assays, we identified a novel Nox2 inhibitor, TG15-132, that prevents PMA-stimulated oxygen consumption and reactive oxygen species (superoxide radical anion and hydrogen peroxide) formation upon acute treatment in differentiated HL60 cells. Long-term treatment with TG15-132 attenuates the induction of genes encoding Nox2 subunits, several inflammatory cytokines, and iNOS in differentiated THP-1 cells. Moreover, TG15-132 shows a relatively long plasma half-life (5.6 h) and excellent brain permeability, with a brain-to-plasma ratio (>5-fold) in rodent models. Additionally, TG15-132 does not cause any toxic effects on vital organs or blood biomarkers of toxicity in mice upon chronic dosing for seven days. We propose that TG15-132 may be used as a Nox2 inhibitor and a potential neuroprotective agent, with possible further structural modifications to increase its potency.

## 1. Introduction

The production of cellular oxidizing, nitrating, and halogenating species, involving the superoxide radical anion (O_2_^•−^), hydrogen peroxide (H_2_O_2_), hypochlorous acid (HOCl), peroxynitrite (ONOO^−^), and the hydroxyl radical (^•^OH), collectively termed reactive oxygen species (ROS), is a common feature of pathological states involving an inflammatory component (Figure 1) [1,2]. Production of ROS by inflammatory cells is essential for the host immune response, but sustained activation of the immune system may lead to damage and dysfunction of the host cells [3]. The primary source of ROS in inflammatory cells is NADPH oxidase-2 (Nox2); therefore, the development of pharmacological approaches to inhibit Nox2 is an active area of research with significant translational potential [4,5]. Numerous rodent models of diseases with an inflammatory component showed beneficial effects following the genetic deletion of Nox2-related proteins, validating Nox2 activity as a potential therapeutic target [6,7,8,9].

Neuroinflammation, defined as an inflammatory response within the brain or spinal cord, is a critical contributing factor to many neurological diseases, ranging from status epilepticus to Alzheimer’s disease, Parkinson’s disease, and amyotrophic lateral sclerosis. Available data indicate a complex relationship between oxidative stress and inflammatory mediators, including both mutual activation and regulatory mechanisms [3,4,10]. 

Due to the high abundance of preclinical and clinical evidence for the involvement of oxidative and/or nitrative stress in inflammatory diseases, clinically relevant antioxidants have been proposed as possible protective agents. Both pre-clinical and clinical data provided mixed results, bringing into question the ability of oxidant scavengers to yield a therapeutic benefit [11,12,13]. Instead of aiming to scavenge the already formed reactive species, a more promising strategy may be to prevent their formation by blocking the activity of the enzymes responsible for the initiation of the ROS cascade (Figure 1) [14,15,16]. 

For the last two decades, there has been an active quest to identify new inhibitors of NADPH oxidases as potential therapeutic agents [17,18,19]. However, the results of these attempts were only moderately successful, and many putative inhibitors have been subsequently shown to act indirectly, owing to significant off-target effects on cell metabolism, or to simply interfere with the Nox2 activity assays [20,21,22]. One of the reasons for this has been the limited number of assays that can specifically measure the products of Nox activity: O_2_^•−^ and H_2_O_2_. In fact, many assays rely on the measurement of a lesser-defined “oxidative burst,” with the probes used relying on peroxidatic activity, typically accompanying increased Nox2 activity. Thus, inhibitors and substrates of peroxidases interfered with peroxidase-catalyzed probe oxidation and were disguised as apparent Nox inhibitors [21,23]. Moreover, most putative Nox inhibitors failed to show Nox-inhibitory activity in cell-free assays [20]. Recent progress in the development of redox probes and a better understanding of the chemistry behind the redox probes allowed for the design of rigorous assays to measure O_2_^•−^ and H_2_O_2_ in cell-based systems, and the establishment of a high-throughput screening-compatible workflow for rigorous identification of the inhibitors of NADPH oxidases [24,25,26]. 

The application of Nox2 inhibitors for neuroprotection requires the compounds to cross the blood–brain barrier (BBB) to achieve therapeutically relevant levels in the brain. One potential strategy is to repurpose existing drugs used in neurological disorders, and several phenothiazine-based antipsychotic drugs have been shown to exhibit Nox2 inhibitory activity in vitro [25,27] and in vivo [28]. A separate approach based on medicinal chemistry-based modification of a Nox2 inhibitor of limited specificity, GSK2795039 [29], recently resulted in the development of a more specific Nox2 inhibitor, NCATS-SM7270, which shows protective effects against traumatic brain injury [30]. 

Here, we report the development of a novel class of BBB-permeable Nox2 inhibitors, characterize their Nox2 inhibitory potency, and determine their effects on the expression of genes associated with the inflammatory response in macrophages. 

## 2. Materials and Methods

### 2.1. Materials

Coumarin-7-boronic acid (CBA) was synthesized as described previously [31], and the 0.1 M stock solution was prepared in DMSO and stored at −20 °C. Amplex Red (10-acetyl-3,7-dihydroxyphenoxazine) was purchased from Cayman Chemicals (Ann Arbor, MI, USA), and the 50 mM stock solution was prepared in DMSO and stored at −20 °C. Hydroethidine (5-ethyl-5,6-dihydro-6-phenyl-3,8-phenanthridinediamine, also known as dihydroethidium) was from Cayman Chemicals, and the 20 mM stock solution was prepared in deoxygenated DMSO, aliquoted, and stored at −80 °C. The standards for hydroethidine (HE) oxidation products were prepared as described previously [32,33]. Horseradish peroxidase (HRP, type VI) from Sigma-Aldrich (Milwaukee, WI, USA) was prepared at a concentration of 1 kU/mL in phosphate buffer (50 mM, pH 7.4) containing 100 µM DTPA and stored at 4 °C. DNA (from salmon testes) was obtained from Sigma-Aldrich and dissolved at a concentration of 2 mg/mL in Tris buffer (10 mM) containing EDTA (1 mM) and stored at 4 °C. The Cell Titer Glo 2.0 cytotoxicity assay kit was obtained from Promega (Madison, WI, USA). Phorbol 12-myristate 13-acetate (PMA) was purchased from Sigma-Aldrich, and the 10 mM stock solution was prepared in DMSO and stored at −20 °C. Diphenyleneiodonium chloride (DPI) was from Cayman Chemicals, and the 10 mM stock solution was prepared in DMSO and stored at −20 °C. Superoxide dismutase from bovine heart, catalase from *Corynebacterium glutamicum*, acetonitrile (HPLC grade), DAPP, and apocynin were obtained from Sigma-Aldrich. Diapocynin was synthesized as described previously [34]. Trifluoroacetic acid was purchased from Thermo Scientific (Waltham, MA, USA).

### 2.2. Cell Culture

Human promyelocytic leukemia HL60 cells (Sigma-Aldrich) were grown in RPMI 1640 medium (Gibco, 11875-093, Billings, MT, USA) supplemented with 10% FBS (Omega Scientific (Tarzana, CA, USA)), 100 units/mL penicillin, and 100 µg/mL streptomycin, and were incubated at 37 °C and 5% carbon dioxide (CO_2_). For differentiation to neutrophil-like cells, HL60 cells were pelleted and resuspended in fresh media containing 1 µM of all trans-retinoic acid and incubated for four to five days. Nox2 activation was achieved by treating differentiated HL60 (*d*HL60) cells with 1 µM PMA [24]. Human leukemia monocytic cells, THP-1, were grown in RPMI 1640 medium + GlutaMAX^TM^ (Gibco, 61870-036) and supplemented with 10% FBS, 10 mM HEPES, and 1 mM sodium pyruvate and incubated at 37 °C in 5% CO_2_ environment. To induce differentiation into macrophages, THP-1 cells were treated with PMA (100 ng/mL) for 48 h. To test the efficacy of Nox2 inhibitors, treatment was performed with or without the inhibitor, along with PMA [35].

### 2.3. Plate Reader-Based Analyses of O_2_^•−^ and H_2_O_2_ Production by Activated dHL60 Cells

Plate reader-based analyses of the production of O_2_^•−^ and H_2_O_2_ by PMA-activated *d*HL60 cells were performed according to a published protocol with minor modifications [36]. Briefly, for the treatment of *d*HL60 cells with different concentrations of the compounds, 0.5 mL deep-well 96-well plates were used to dispense and store aliquots of their 10 mM DMSO solutions. Using a Multidrop™ PicoIT dispenser (Thermo Scientific), different volumes of the solutions of candidate Nox inhibitors (10 or 50 mM in DMSO) were injected into the bottom of 0.5 mL V-bottom 96-well plates; then, they were sealed with aluminum foil (VWR, Batavia, IL, USA) using a semiautomatic thermal sealer (BT LabSystems, St. Louis, MO, USA) and stored at −20 °C to reduce freeze-thawing the solutions. The plates with inhibitors were allowed to come to room temperature before the addition of the cell suspensions. *d*HL60 cells were pelleted and resuspended in HBSS (Invitrogen, Waltham, MA, USA) supplemented with 25 mM HEPES (pH 7.4) and 100 µM DTPA to a cell density of 2 × 10^5^ cells/mL. *d*HL60 cells were added to the V-bottom plates at a volume of 210 µL/well. Next, cells were incubated with inhibitors for 30 min at 37 °C in a CO_2_-free incubator under ambient oxygen and then transferred to black, clear-bottom 96-well plates (200 µL/well) after incubation. Freshly prepared PMA plus probe solutions in HBSS (5× concentrated, 50 µL/well) included PMA (5 µM) with CBA (500 µM) or with Amplex Red (250 µM) plus HRP (0.5 U/mL) for H_2_O_2_ detection, or with HE (50 µM) plus DNA (0.5 mg/mL) for O_2_^•−^ detection. Once probes + PMA were added, plates were sealed with sealing film and immediately placed in the pre-warmed plate readers (37 °C). Fluorescence intensities were monitored kinetically from the bottom of the plates every 1.3 min for a period of 2 h using Beckman Coulter (Brea, CA, USA) DTX880 and BMG Labtech (Ortenberg, Germany) FLUOstar Omega plate readers. The fluorescence excitation and emission filters used are listed in Table 1. DMSO alone and DPI (10 µM) were used as negative and positive controls, respectively. At the concentration used (10 µM), DPI did not show any cytotoxic effects, as measured by the Cell Titer Glo 2.0 cytotoxic assay, under the conditions used. The IC_50_ values were determined using at least five independent measurements, each consisting of technical quadruplicates per concentration.

### 2.4. Rapid HPLC Analyses of O_2_^•−^ Production by Activated dHL60 Cells

For high-performance liquid chromatography (HPLC)-based analyses of superoxide production by activated *d*HL60 cells, the cells (0.8 mL, 2 × 10^5^ cells/mL) were pretreated with inhibitors for 30 min, as described above, followed by the addition of 200 µL of the mixture of PMA (5 µM) and HE (50 µM); then, they were incubated for 1 h at 37 °C in the dark. Next, 0.5 mL aliquots of the cell suspensions were transferred into 1.5 mL microcentrifuge tubes prefilled with 10 µL of a mixture of 3,8-diamino-6-phenylphenanthridine (50 µM, used as an internal standard), superoxide dismutase, and catalase (5 mg/mL and 5 kU/mL, respectively, used to stop probe oxidation), and the tubes were centrifugated (5 min, 3000× *g*) to pellet the cells. Clear supernatants (0.2 mL) were transferred into HPLC vials and analyzed for HE, 2-hydroxyethidium (2–OH–E^+^), ethidium (E^+^), and diethidium (E^+^–E^+^) content, as described previously [37]. Briefly, an Agilent 1100 HPLC system equipped with an Ascentis Express Phenyl-Hexyl column (50 mm × 4.6 mm, 2.7 µm, Supelco, Bellefonte, PA, USA) and absorption and fluorescence detectors was used. The analytes were eluted in isocratic mode using a mobile phase consisting of 65% water, 35% acetonitrile, and 0.1% trifluoroacetic acid at a flow rate of 2 mL/min. Samples (25 µL) were injected, and the products were detected by monitoring absorbance (290 and 370 nm) and fluorescence (exc. 370 nm, emi. 565 nm) signals. 

### 2.5. Oxygen Consumption Measurements

Oxygen consumption in *d*HL60 cells was monitored before and after the sequential addition of the candidate Nox inhibitors and PMA using a Seahorse XFe96 extracellular flux analyzer (Agilent Technologies, Santa Clara, CA, USA) [24,36]. The cell suspension was prepared in phenol red-free RPMI medium (without bicarbonate) and aliquoted (80 µL/well) into a 96-well plate to obtain a final cell number of 2 × 10^4^ cells/well. Cells were spun down and an additional amount of RPMI medium (100 µL/well) was added gently so as not to disturb the cells at the bottom of the wells. The solutions for injections were prepared 10× concentrated, loaded into the injection ports of the Seahorse XFe96 cartridge, and the volumes injected were 20 µL and 22.2 µL for the compounds tested and PMA, respectively. The oxygen consumption rates were monitored for 30 min before injection (baseline respiration rate), 60 min after injection of different concentrations of the candidate Nox inhibitors, and 2 h after injection of PMA (final concentration: 1 µM). Increases in OCR values after PMA injection were used as a measure of Nox2 activity.

### 2.6. Cytotoxicity

The cytotoxicity of the inhibitors was tested by measuring the amount of ATP in the wells containing PMA-stimulated *d*HL60 cells and the candidate inhibitors. Cells were prepared as described for the plate reader-based measurements for H_2_O_2_, except that the cells were maintained with the inhibitors in a CO_2_-free incubator at 37 °C for 2.5 h to mirror the conditions during the fluorescence-based monitoring of H_2_O_2_ described above. After incubation, cells (100 µL/well) were transferred to white 96-well plates (Greiner, Kremsmünster, Austria), and the total cellular ATP amount was determined using the Cell Titer Glo 2.0 (Promega) cytotoxicity assay (100 µL/well) by measuring the luminescence intensity using a Beckman Coulter DTX880 plate reader. 

### 2.7. Gene Expression Analysis

mRNA expression analysis was performed as previously described [38,39,40]. Briefly, THP-1 cells were treated with PMA (100 ng/mL) with or without the Nox2 inhibitor TG15-132 (3.0 µM) for 48 h. RNA extraction was done using the Quick-RNA miniprep kit (cat. no. R1055, Zymo Research, Irvine, CA, USA) and followed by cDNA synthesis using the qScript cDNA superMix kit (cat. no. 84035, Quantabio, Beverly, MA, USA). Quantitative real-time PCR was performed using specific primers designed to amplify target genes (Table 2). Data obtained were analyzed and converted into fold changes in expression with respect to vehicle-treated cells.

### 2.8. Plasma and Brain In Vivo Pharmacokinetics Studies 

TG15-132 at a 20 mg/kg dose in a vehicle formulation of 10% DMSO, 50% PEG400, and 40% reversed osmosis-purified water (*v*/*v*) was administered to healthy male C57BL/6 mice weighing an average of 27 g by intraperitoneal (i.p.) injection. Blood samples (approximately 60 μL) were collected from the retro-orbital plexus of three animals at times 0.08, 0.25, 0.5, 1, 2, 4, 8, and 24 h after compound administration. Along with the terminal blood samples, the brain samples were also collected at the time points 0.25, 0.5, 2.0, 4.0, 8.0, and 24 h. 

Blood samples were collected into labeled microtubes containing dipotassium EDTA solution (20%) as an anticoagulant. Plasma was immediately harvested from the blood by centrifugation at 4000 rpm for 10 min at 4 ± 2 °C and stored below −70 °C until bioanalysis. After isolation, brain tissue samples were rinsed three times in ice cold normal saline (for 5–10 s/rinse using ~5–10 mL normal saline in disposable petri dishes for each rinse) and dried on blotting paper. Tissue samples were homogenized using ice-cold phosphate-buffered saline (pH 7.4). The total homogenate volume (µL) was three times the tissue weight (mg). All homogenates were stored below −70 ± 10 °C until bioanalysis. Concentrations of TG15-132 in plasma and brain samples were determined by fit-for-purpose liquid chromatography with the tandem mass spectrometry method (Sai-Life, Hyderabad, India). Phoenix WinNonlin (version 8.0, Certara, Princeton, NJ, USA) was used to analyze the pharmacokinetic parameters. 

### 2.9. Short-Term In Vivo Toxicity

All animal experiments were performed according to protocols and guidelines approved by the Emory University Animal Care and Use Committee. The in vivo toxicity of TG15-132 was assessed using C57BL/6 mice (cat. no. 027, Charles River Laboratories, Wilmington, MA, USA). Animals were injected daily (i.p.) with saline, vehicle (DMSO 10%, PEG400 30%, water 60%), or 20 mg/kg of TG15-132 in vehicle for seven days, with body weight recorded each day. At the end of treatment, organs were harvested to study organ weight loss, and blood was collected from each mouse via cardiac puncture, and serum was isolated. Serum was analyzed for kidney function using blood urea nitrogen and creatinine, liver function using alanine amino transferase, and tissue damage using lactate dehydrogenase [41]. 

### 2.10. Synthesis of Nox2 Inhibitors TG15-132 and Derivatives

Previously synthesized and reported molecules, 3-(3-(dimethylamino)propyl)-2-thioxo-2,3-dihydroquinazolin-4(1H)-ones 1 and 2, were subjected to alkylation with commercially available compounds 3-(2-bromoethyl)-1H-indole (4) or 2-(2-bromorethylpyridine) (3) in the presence of potassium carbonate base to provide TG15-132, TG15-139, TG15-131, and TG15-124 in moderate yields (Figure 1). These free bases were treated with an aqueous HCl solution to provide the corresponding hydrochloride salts. Free bases were dissolved in DMSO for the determination of the direct Nox2 inhibitory effects of the compounds, but similar effects were observed for both free bases and hydrochloride salts. In most other experiments, the hydrochloride salts of the synthesized inhibitors were used.

Similarly, four other derivatives, TG17-55, TG17-56, TG17-57, and TG15-293, were synthesized from fluoro-isatoic anhydride (5) via precursors (**8a**, **8b**, and **11**), as shown in Figure 1 and Figure 2. All novel compounds were characterized by nuclear magnetic resonance, liquid chromatography-mass spectrometry, and HPLC methods (Appendix A).

## 3. Results

### 3.1. Identification of Novel Nox2 Inhibitors

In an ongoing quest to identify and develop Nox2 selective inhibitors for therapeutic use, we previously conducted high-throughput screening (HTS) on a library of >200,000 compounds using a cell-free assay and found only ebselen (selenium compound) and its sulfur congeners as modest Nox2 inhibitors [42]. Subsequently, we re-screened approximately 50 compounds from our synthetic library, including some widely used (in the literature), commercially available putative Nox2 inhibitors (DPI, apocynin, diapocynin, and others) to find genuine Nox2 selective inhibitors [24]. From these, we identified TG15-132 (Figure 2) as a genuine Nox2 inhibitor, and a few other derivatives synthesized around this compound also showed low micromolar Nox2 inhibitory activity, as detailed below.

### 3.2. Characterization of Nox2 Inhibition in Differentiated HL60 Cells

We used HL60 leukemia cells differentiated with all-trans retinoic acid to neutrophil-like cells (*d*HL60) as a model to screen and characterize potential Nox2 inhibitors. Upon treatment of *d*HL60 cells with PMA, the NADPH oxidase cytosolic and membrane protein partners assemble to form a fully active complex that produces the short-lived O_2_^•−^, which is subsequently dismutated to a more stable product, H_2_O_2_. Thus, monitoring the changes in H_2_O_2_ produced by activated *d*HL60 cells provides a convenient and high-throughput screening-compatible way to study the potency of candidate Nox2 inhibitors [24,25,36]. We used CBA to monitor H_2_O_2_ production, as boronate compounds react directly with H_2_O_2_ without the requirement of peroxidase, thus minimizing the potential interference of peroxidase substrates and inhibitors with the assay. During the reaction, CBA was converted to a blue fluorescent product, 7-hydroxycoumarin (COH, also known as umbelliferone, Figure 3A). Among the inhibitors tested, TG15-132 and TG15-139 were the most promising, with half-maximal inhibitory concentration (IC_50_) values of 4.5 and 3.0 µM, respectively. Representative traces of CBA oxidation and titration curves for both inhibitors are shown in Figure 3B,C. To exclude the possibility that the results obtained were due to potential cytotoxicity of the tested compounds, the inhibition studies were accompanied by measurements of cell viability. As shown in Figure 3D,E, both compounds show Nox2 inhibitory potency (blue squares) at concentrations at which no decrease in cell viability (red circles) was observed.

As an orthogonal assay of H_2_O_2_ production, we used peroxidase-catalyzed oxidation of Amplex Red to red fluorescent resorufin (Figure 4A), one of the most widely used assays to assess extracellular H_2_O_2_. Consistent with the data obtained with the CBA probe, we observed dose-dependent inhibition of Amplex Red oxidation (IC_50_ = 7.9 and 5.0 µM for TG15-132 and TG15-139, respectively), confirming the ability of the compounds to block H_2_O_2_ production at low micromolar concentrations (Figure 4). 

The IC_50_ values for all compounds showing inhibitory activity, determined using both H_2_O_2_ assays and the viability assay, are presented in Table 3. For comparison, we also tested DPI, a general flavoprotein inhibitor, and apocynin and its peroxidase-catalyzed oxidation product, diapocynin, in our study. Apocynin is widely used as an inhibitor of NADPH oxidase, and diapocynin was proposed to be its metabolite responsible for its inhibitory activity. However, these compounds did not show Nox2 inhibitory activity when tested at concentrations of up to 50 µM, in accordance with recent reports. DPI (10 µM) consistently showed inhibitory activity and was used in all experiments as a positive control.

### 3.3. Structure–Activity Relationship Studies

After conducting a limited number of structure–activity relationship studies on this class of compounds, we observed that replacing the indole moiety (connected to sulfur with an ethylene linker [TG15-132 and TG15-139]) with a pyridine moiety (see TG15-131 and TG15-124 in Figure 1) resulted in a significant loss in Nox2 inhibitory potency (not shown). Likewise, replacing the same indole-moiety with a much simpler amino-alkyl chain (TG17-55, Figure 1) also eliminated potency. We then wondered whether keeping the indole moiety intact while modifying the N,N-dimethylaminopropyl moiety (other region) would retain the potency. To answer this question, we synthesized two compounds, TG17-56 and TG17-57, as shown in Figure 2. Interestingly, these derivatives showed low micromolar potencies (5.3 µM and 7 µM, Table 3), suggesting that this region is amenable for structural modification. Furthermore, we have also synthesized a derivative, TG15-293, which bears an extra carbonyl group in the structure in comparison to TG17-57. This compound showed about 3-fold less potency than TG17-57 in the CBA assay and about 7-fold less potency in the Amplex Red assay (Table 3). Future studies will expand this region further, and the core quinazoline for structural modification. 

Comparison of the H_2_O_2_ production inhibitory potency and cell cytotoxicity of the tested candidate inhibitors led to the selection of the TG15-132 compound as the lead hit, showing high potency and low cytotoxicity. This compound was, therefore, advanced for additional confirmatory assays, including the measurements of the effect of TG15-132 on PMA-stimulated O_2_^•−^ production and O_2_ consumption by *d*HL60 cells.

We used hydroethidine (HE) to monitor O_2_^•−^ production. In the presence of O_2_^•−^, HE is converted to a O_2_^•−^-specific red fluorescent product, 2–OH–E^+^ (Figure 5A), while other oxidants may form another fluorescent product, E^+^, and dimeric products, including E^+^–E^+^ [43,44]. The fluorescence intensity of 2–OH–E^+^ and E^+^ can be further increased by binding to DNA; thus, the assay medium was supplemented with DNA for plate reader-based monitoring of probe oxidation. Stimulation of *d*HL60 cells with PMA led to a time-dependent increase in the fluorescence intensity, which was dose-dependently inhibited by cell pretreatment with TG15-132 (Figure 5B,C). To test whether the fluorescence signal was due to superoxide-dependent probe oxidation, the plate reader-based analyses were accompanied by rapid HPLC analysis of the assay media after 1 h of incubation. Oxidation of HE was accompanied by the formation of 2–OH–E^+^ as the major product, and both the extent of HE consumption and 2–OH–E^+^ formation were inhibited by cell pretreatment with TG15-132 in a dose-dependent manner (Figure 5D,E). This confirms that the compound blocks O_2_^•−^ production by activated *d*HL60 cells.

Further confirmation of the Nox2 inhibitory activity of TG15-132 was obtained by measuring the rates of PMA-stimulated oxygen consumption. We used the Seahorse XFe96 Extracellular Flux Analyzer, which enables repeated measurements of oxygen consumption rates and sequential injections of up to four different treatments during measurements, eliminating the need for chemical probes. As shown in Figure 6, the injection of PMA resulted in an immediate increase in the oxygen consumption rate, which represents the oxidative burst due to Nox2 activation. The injection of candidate inhibitors prior to PMA injection enabled monitoring of Nox2 inhibitory activity as well as potential mitochondrial-inhibitory activity. Upon establishment of the stable baseline, the compound was injected, and oxygen consumption rates were monitored for 1 h before activation of Nox2 by PMA. There was no inhibition of basal respiration at concentrations at which the compound showed significant inhibition of PMA-induced OCR. This further confirms that TG15-132 inhibits Nox2 activity in PMA-stimulated *d*HL60 cells without affecting mitochondrial function and viability.

### 3.4. Effect of TG15-132 on Molecular Markers of Inflammation in THP-1 Cells

The relationship between redox stress and inflammation involves cytokine-dependent activation of pro-inflammatory genes and increased expression of NADPH oxidase components. Previous reports suggest that putative NADPH oxidase inhibitors may also inhibit the protein expression of NADPH oxidase components, which may contribute to the effects observed [45,46]. While the data obtained using *d*HL60 cells (described above) involved acute treatment and inhibition of Nox2 activity, we used human THP-1 cells differentiated to express a macrophage phenotype to study the effect of the selected Nox2 inhibitor, TG15-132, on the expression of Nox2 components and other proinflammatory genes. THP-1 cells were treated with PMA (100 ng/mL) for 48 h, with or without TG15-132 (3 µM), followed by mRNA extraction and analysis of gene expression. Treatment with PMA led to increased expression of *gp91^phox^*, *p47^phox^*, *p67^phox^*, and *IL-1β*, whereas expression of *TNF* was significantly reduced (Figure 7A). TG15-132 treatment reduced the expression of *gp91^phox^*, *p47^phox^*, *p67^phox^*, *IL-1β*, and *TNF* compared with cells treated with PMA alone. In an additional experiment, TG15-132 was evaluated against the effects of GSK 2795039, a recently reported Nox2 inhibitor (Figure 7B). Under the same experimental conditions, TG15-132 treatment led to a statistically significant reduction in the expression of *p47^phox^*, *IL-1β*, and *TNF*, whereas treatment with GSK 2795039 (3 µM) did not reduce the expression of Nox2 components and inflammatory mediators, and the expression of *TNF* was increased compared with PMA-treated cells. 

### 3.5. In Vivo Pharmacokinetic and Toxicity Study of Seven Day Dosing

Rapid metabolism and the inability to cross the BBB represent major obstacles in the application of bioactive agents in neurodegenerative diseases. Based on the premise that Nox2 plays a major role in oxidative damage and exacerbates neurodegenerative diseases, we envision applications of brain-permeable Nox2 inhibitors in mechanistic studies in vivo, and, long-term, in the development of novel therapeutic strategies. Therefore, we performed in vivo pharmacokinetics studies to determine the time profile of plasma concentration and accumulation of TG15-132 in the brain tissue of Sprague Dawley rats and C57BL6 mice. As shown in Figure 8, TG15-132 displayed a plasma half-life of 3.7 h and a brain half-life of 5.6 h after a single dose of 20 mg/kg by i.p. injection in Sprague Dawley rats, with an AUCinf of 10,180 (h∙ng/mL) and clearance rate of 32.8 (mL/min/kg). Interestingly, the brain-to-plasma ratio for this compound is exceptionally high (>10-fold up to 8 h after administration in Sprague Dawley rats) (*n* = 2 technical and *n* = 2 experimental repeats). Similarly, C57BL6 mice that were administered a single dose of 20 mg/kg TG15-132 (i.p. injection) displayed a plasma half-life and brain half-life of 2.7 h, with a brain-to-plasma ratio >6-fold, indicating that TG15-132 is a highly suitable compound for proof-of-concept studies and a novel chemical scaffold useful for optimizing and enhancing the potency of this class of Nox2 inhibitors. Likewise, TG15-139 displayed exceptionally high brain-penetration with a brain-to-plasma ratio >23-fold (not shown), suggesting that high brain penetration is not restricted to a single compound in this class. 

We found the pharmacokinetics of these compounds to be impressive, so we wanted to determine whether TG15-132 would show any in vivo toxicity to the animals before we advanced to potential proof-of-concept studies in preclinical animal disease models. Considering the observed pharmacokinetic profile and efficient accumulation of TG15-132 in the brain (Figure 8), we envisioned that a week-long treatment may be sufficient to result in inhibition of both Nox2 activity and inhibition of the expression of inflammatory markers. Therefore, we selected a seven-day administration period to determine the potential toxicity of TG15-132. As shown in Figure 9, seven days of daily administration of TG15-132 (20 mg/kg) caused only a marginal and nonsignificant loss of body weight (7%), and no changes to the weights of the lung, liver, or brain. We also analyzed the serum to look for kidney, lung, and tissue injury markers. We observed no differences between mice treated with vehicle vs. TG15-132 (*n* = 6). TG15-132 treatment led to a significant drop in blood urea nitrogen compared with saline-treated mice.

## 4. Discussion

The key findings presented in this manuscript include the identification of a novel and relatively potent Nox2 inhibitor with an IC_50_ at low micromolar concentrations and favorable cytotoxic and in vivo safety characteristics. The presented data provide a foundation for a class of Nox2 inhibitors that can be used as basic chemical scaffolds to develop future, more potent Nox2 inhibitor candidates. The described compounds exhibit potency in the low micromolar range, which is multifold lower than the widely used experimental NADPH oxidase inhibitor apocynin. The determined inhibitory potency in the low micromolar region may enable targeting of various diseases that have shown involvement of NADPH oxidase dysregulation as either a core cause or a secondary effect in the pathogenesis. Significant efforts are needed to optimize and improve upon our discoveries to develop potential candidate compounds that are more potent and have better properties. 

Oxidative stress has been associated with the initiation and progression of various neurological disorders [47,48,49]. Furthermore, oxidative stress is linked to alterations in the inflammatory response in models of these diseases [47,50,51,52]. Therefore, studying the relationship between NADPH oxidase inhibition and the alteration of inflammatory mediators in response to treatment with NADPH oxidase inhibitors is key for a thorough understanding of the mechanism(s) of action of the inhibitors. Not only did we observe inhibition of Nox2 activity and a significant reduction in the expression of key subunits of the NADPH oxidase complex with the developed inhibitor (TG15-132), we also observed a reduction in the levels of inflammatory markers in the cell model used. Therefore, this new class of Nox2 inhibitors is promising for applications where inhibition of markers of oxidative stress and inflammation is desired, and for the development of novel drug candidates targeting NADPH oxidase and inflammation in diseases.

When it comes to targeting neurological disorders, the main roadblock faced by any potential candidate is permeability across the BBB to reach the central nervous system. TG15-132 shows very promising permeability across the BBB, with a brain-to-plasma ratio staying above the 10-fold mark for at least 8 h post injection. This observation further supports TG15-132 as a potential candidate for targeting NADPH oxidase in vivo in models of neurodegenerative diseases. 

Although the described Nox2 inhibitor candidate shows significantly better characteristics compared with many currently available Nox2 inhibitors, its use as a selective Nox2 inhibitor and mechanistic tool requires further research, including models for other NADPH oxidase isoforms and cell-free Nox2 assays to decipher the mechanism of inhibition. The presented data suggest that both direct interaction of TG15-132 with NADPH oxidase subunits and transcriptional regulation of the expression of Nox2 subunits may be involved. It seems unlikely that transcriptional regulation is responsible for the inhibitory activity observed in *d*HL60 cells, as preincubation with the inhibitors in all the activity assays lasted 30 min, while the effects on the expression of the transcripts of NADPH oxidase-2 subunits were observed after 48-h incubation. Whether there is a direct link between these two activities remains to be established. In addition, while we presented initial structure-activity relationship data that show the motifs responsible/required for the inhibitory activity observed, rigorous structural modification studies are needed to optimize the structure of candidates with optimal physical and functional characteristics. Furthermore, before application of the identified inhibitor for neuroprotection, additional testing of its effect on the function of microglial and neuronal cells is needed to extend the observed Nox2 inhibitory activity and low cytotoxicity to brain-specific cell types. 

In summary, TG15-132 and related compounds are a promising class of bioactive compounds that are candidates for targeting NADPH oxidase in models of neurodegenerative diseases. TG15-132 shows a Nox2 inhibitory potency superior to numerous less potent and nonspecific inhibitors currently being used (e.g., apocynin). Furthermore, TG15-132 has the ability to modulate the transcription of NADPH oxidase subunits gp91^phox^, p47^phox^, and p67^phox^, as well as inflammatory mediators, including IL-1β. Finally, TG15-132 shows highly desirable BBB permeability, with an exceptional brain-to-plasma ratio and significantly longer retention in the brain. Collectively, all of the features of TG15-132 point to it being a highly promising therapeutic agent to target neurological diseases where their etiology and/or pathogenesis is directly or indirectly related to NADPH oxidase dysregulation. 

## 5. Conclusions

The need for reliable Nox2 inhibitors for biomedical research compelled us to develop a distinct class of Nox-selective inhibitors. In this study, we report a novel quinazoline class of compounds as Nox2 inhibitors. These compounds display low micromolar potency, high brain penetration, and anti-inflammatory properties in cellular models in vitro, and may be suitable compounds for preclinical studies using in vitro and in vivo disease models. 

## Data Availability

The data presented in this study are available in the figures and tables included within the manuscript and the Appendix A.

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
