# Peer review of "Novel NADPH Oxidase-2 Inhibitors as Potential Anti-Inflammatory and Neuroprotective Agents"

_antioxidants, 2023, doi:10.3390/antiox12091660_

Round 1

Reviewer 1 Report

The search for novel and specific NADPH oxidases inhibitors is a hot issue in the field for their putative utility as therapeutic tools. In this manuscript Juric et al. describe the development a novel NOX2 selective inhibitors. The evidence provided is consistent with the notion sustained by the authors regarding the effectiveness of the novel inhibitors against NOX2.

The discovery of novel NOX2 inhibitors is without doubt interesting to the field, so therefore this manuscript is relevant.

Please find below my comments.

Specific comments:

1.    As a system to monitor NOX2 activity and the effect of the inhibitor here described the authors use differentiated HL60 cells, which mainly express NOX2. How are the authors sure that the inhibitors here described do not affect other NOX isoforms?. If the authors claim that TG-15-132 and TG-15-139 specifically inhibit NOX2 activity, evidence showing that they do not affect other NOX isoforms should be provided.

2.    The IC50 of TG-15-132 and TG-15-139 deduced from figures 3 and 4 are slightly different, even when in both assays H2O2 is detected. Please explain.

3.     In figure 7 the authors show how TG-15-132 negatively affects to the expression of several components of the NOX2 complex, which could also contribute to the inhibition of NOX2-driven ROS production. This make me wonder what is more important for TG-15-132 mechanism of action: inhibition of enzyme activity or the control of gene expression? Please elaborate.

Reviewer 2 Report

Review comments are attached

Round 2

Reviewer 1 Report

The authors have addressed all the comments satisfactorily

Reviewer 2 Report

The present form of the revised manuscirpt is  satisfactory